# The impact of work-family conflict on job burnout among community social workers in China

**Dian Song[1], Jie Zhao[1], Hainan Wu[2]\*, Xueyi Ji[2]\***

1 School of Political and Public Administration, Soochow University, Suzhou, China, 2 School of Finance and Public Administration, Anhui University of Finance & Economics, Bengbu, China

\* wuhainan@aufe.edu.cn (HW); 2043872326@qq.com (XJ)

## Abstract

In China, for community social workers, work-family conflict has become a common phenomenon that may harm their well-being. Based on the analysis of a survey of community social workers in four cities in China, this study demonstrated that community workers' work-family conflict significantly affects burnout, role overload mediates the relation between work-family conflict and burnout, and cognitive crafting negatively moderates the relation between role overload and burnout. The conclusions validate the job crafting theory and enrich the research on job burnout under the JD-R model. The practical significance of the study is that on the one hand, community and individual workers can effectively alleviate burnout by clarifying their roles. On the other hand, it also reminds managers that they should pay attention to the physical and mental health of social workers to enable them to develop healthily.

## 1. Introduction

Community social workers play an essential role in community development [1]. With the development of urbanization, Chinese communities are increasingly reliant on community social workers to meet the needs of residents, and the number of community social workers is growing [2]. In China, the saying "a thousand lines above, a needle below" is often used to describe the busyness of community social workers. Line refers to the various tasks that the community needs to face, needle refers to the community social worker. Due to the complexity and variety of their job, community social workers often need to work overtime. Research has shown that a significant proportion of community social workers suffered from depression [3], and severe burnout [4], especially during the COVID-19 pandemic [5,4]. Burnout leads to high turnover rates in community social workers, which in turn results in the decline of service quality [2]. Research has shown that socio-demographic factors, role stress [6], prosocial personality [7], job demands [8] and work overtime [9] influence the burnout of community social workers.

Although the above factors have a significant impact on job burnout, they overlook the fact that community workers often work overtime [9,10]. High complexity and intensity of the work often makes it difficult for community workers social to balance family life, which in turn leads to work-family conflicts [11,12]. Work-family conflict occurs when the

**Data availability statement:** All relevant data are within the manuscript and its Supporting Information files.

**Funding:** This study is a phased achievement of the Humanities and Social Sciences Key Project in Anhui Province (2022AH050570), funded by the Anhui Provincial Department of Education. The funders had no role in study design, data collection and analysis, decision to publish, or preparation of the manuscript.

**Competing interests:** The authors have declared that no competing interests exist.

demands of work/family make individual difficult to play the work and family roles at the same time [13]. However, few researchers have explored how or when work-family conflict impacts the burnout of community social worker. Essentially, work-family conflict is the individual's role overload [14,15]. To deal with role overload, individuals always take job crafting measures to low the effects of role overload [16]. Therefore, drawing upon job demand-resource model, our research takes role overload as mediator and job crafting as moderator to explore the impact of work-family conflict on the burnout of community social workers. The research may contribute the literature in three ways. First, it examines the antecedents of job burnout from the perspective of the work-family relationship, which constitutes a two-way interaction after combining work and family dimensions, enriching the study of work-family conflict. Second, the introduction of role overload and cognitive crafting further enriches the study of the interaction process in the work-requirement-resource model. Third, by unveiling the mechanism of work-family conflict on job burnout, our study provides a theoretical basis and practical guidance for preventing job burnout among community social workers.

## 2. Literature review

### 2.1. Work-family conflict

The earliest research on work-family conflict originated from role theory [17], which regarded the work-family conflict as conflict and stress caused by the incompatibility between the demands of the work and life domains [18]. The source of work-family conflict lies in the degree of incompatibility between role positions. Work-family conflict consists of two directional dimensions: work interferes with family and family interference at work [19,20]. Research has shown that personal characteristics, work domain, and family factors are the major antecedents of work-family conflict [21,22]. Family situation, family commitment, and family-level stress are the drivers of work-family conflict [23]. At the individual level, gender, self-compassion and other factors impact work-family conflict [24]. Most research focuses on the work factors that influence work-family conflict. Scholars have found that work overtime [25], work pressure [26], job entitlement [27], and income [28] affect work-family conflict.

### 2.2. Job burnout

Job burnout is a long-term reaction to the inability to deal with the continuous pressure at work [29]. It consists of three dimensions: emotional exhaustion, depersonalization, and personal accomplishment. Emotional exhaustion refers to the depletion of personal emotional resources to the point of losing energy and enthusiasm at work. Depersonalization refers to the personification and the emergence of cold emotions at work. Personal accomplishment reduction refers to the lack of accomplishment when self-assessment is made in interpersonal relationships at work. In the Shirom-Melamed Burnout Model (S-MBM), job burnout is viewed as an individual's affective state that manifests as a feeling of exhausted physical, emotional, and cognitive energy [30,31]. During the past 40 years, numerous scholars have addressed the antecedents of job burnout from different perspectives. Because of its negative effects, job burnout has received widespread attention. It negatively relates to mental health [32], work engagement [33], and quality of working life [34]. Research has shown that emotional labour [35], work stress or workload [36], role conflict [37], and positive and negative Traits [38] affect job burnout. Regarding work-family conflict, some scholars argued that it can significantly affect the tendency to burnout [39,40]. However, the influence mechanism of work-family conflict on job burnout among community social workers is still unclear.

## 2.3. Role overload

The concept of role conflict was first introduced by [17]. They viewed it as a prevalent and complex form of conflict, a combination of person-role conflict and inter-sender conflict. Gender is a critical factor that influences role overload. Compared with men, they not only have to deal with the work task, but also entail household task [41,9]. Additional, leader-member exchange [42], work-role overload [43], high performance work systems and psychological empowerment [44] affect role overload. Few scholars have explored the relation between role overload and work–family balance [45]. The research on the outcomes of role overload mainly focuses on organizational commitment, job performance, subjective physical and mental health [46], creative behaviour [47], and employee's proactive behavior [48]. Overall, there has been no research exploring the issue of role overload among community social workers and its impact.

## 2.4. Cognitive crafting

Job crafting theory suggests that employees can make their interests, motivations, and passions more compatible with their work, which in turn inspires a range of positive behaviors that change task and relationship boundaries [49]. Job crafting categories into task crafting, relationship crafting, and cognitive crafting. Task crafting refers to changing the physical boundaries of the number and type of tasks; relationship crafting refers to changing the boundaries of the relationships involved in the work. Cognitive crafting refers to changing the cognitive boundaries of work tasks. Cognitive crafting is the most important one which logically precedes task and relationship crafting [50]. For community social workers, there is little room in changing task and relationship. Hence, cognitive crafting is his or her primary choice. Prior Studies mainly focus on the outcomes of job crafting, such as well-being [51], mental health [51], few researchers have explored the moderation role in predicting individual behavior.

## 2.5. Job demands-resources model

The job demands-resources model (JD-R Model) is a representative theory explaining Job burnout and its process and mechanism [52]. JD-R argues that job characteristics can be divided into demand and resource factors. The former refers to the factors that cause work stress, the latter refers to the factors that alleviate work stress (job resources). Job resources play a positive role in relieving stress, and job resources play a buffering role against job demands and burnout. JD-R Model points out that there are two paths of loss and gain for employees in work, demand leads to loss, resource leads to gain [53]. JD-R Model also posits that work resources can buffer the loss of high workload employees. JD-R Model deduces that employees in challenging environments can better transform high work resources into high-level job performance [54].

According to JD-R Model, work-family conflict and role overload are the representatives of job demand, they will impact the burnout of community social worker. Meanwhile, as individual's cognitive resources, cognitive crafting can buffer the impact of the job demand on burnout.. therefore, we proposed the conceptual model(shown in Fig 1):

## 3. Hypotheses development

### 3.1. Work-family conflict and Job Burnout

According to the definition of work-family conflict, work-family interference refers to the interference of working hours, stress, and negative emotions with daily family life, such as a

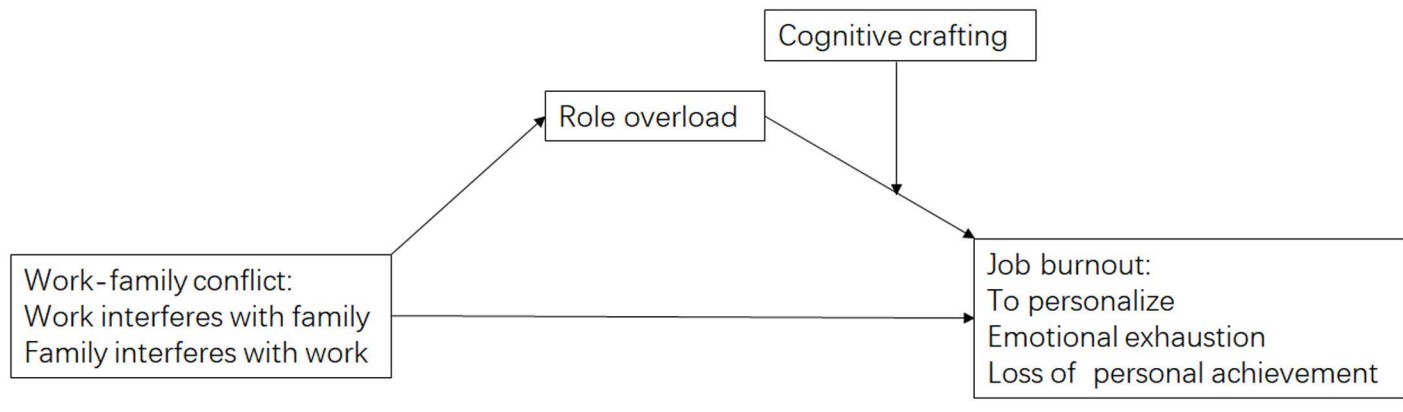

**Fig 1. The conceptual model.**

prolonged lack of family education for children or lack of energy to take care of the elderly due to excessive overtime work. Meanwhile, the family interference dimension mainly refers to family demands, just as excessive family demands can also interfere with work duties. Both interferences promote community social worker to worker overtime, requires them to invest more energy into the work, and puts higher demands on their abilities. Based on the principle of loss and gain in JD-R model, excessive job demands causes in a loss of energy, which in turn leads to job burnout [52]. It will bring individuals experience feelings such as physical fatigue, psychological exhaustion, low achievement [55]. Previous studies have shown that work-family conflict affects the burnout of correctional staff [56], university faculty [57], doctors [58]. Therefore, we propose the following hypotheses:

*H1: Work-family conflict positively relates to job burnout.*

*H1a: Work interference with family positively relates to deindividuation, emotional exhaustion, and personal achievement reduction.*

*H1b: Family interference in work positively relates to deindividuation, emotional exhaustion, and personal achievement reduction.*

### 3.2. Mediating role of role overload

Role overload is the perception caused by an individual's lack of resources to meet job demands [59], stemming from the time [60] and resource pressure [61] individuals face when playing multiple roles. Essentially, work-family conflict is a form of incompatible inter-role conflict in the work and family domains [18]. For community social workers, he or she often engages in interdomain transitions [62]. On the one hand, he needs to frequently undertake urgent and important projects, work overtime frequently, and spend a lot of time on work, which will squeeze out his role in the family. On the other hand, if he invests a lot of energy and time into his family, he will not effectively carry out urgent and heavy work tasks in the community which squeeze his role in work [63]. Therefore, the conflict between two roles will lead to the perception of role overload among community social workers.

Meanwhile, due to the numerous role expectations, individuals are faced with both time and resource pressures caused by role overload [64]. Excessive pressure can reduce employees' ability to control the work environment, make for individuals to have not sufficient time and resources to solve work problems, and leads to fatigue, emotional exhaustion and feeling of energy loss, which in turn leads to job burnout [65]. Combining the relationships among work-family conflict, role overload and job burnout we propose:

*H2: Role overload mediates the relationship between work-family and job burnout.*

*H2a: Role overload mediates the relationship between work interfering with family and burnout.*

*H2b: Role overload mediates the relationship between family interference with work and burnout.*

### 3.3. Moderating role of cognitive crafting

Cognitive crafting is a kind of cognitive-attitudinal changing of job crafting, which precedes relationship crafting and task crafting, stimulates employees' intrinsic motivation to improve their work, positively influences their behavior, improves psychological state, and effectively reduces burnout [50]. Research has found that workers with the highest scores on job crafting experience the highest level of well-being, which can result in an high level of personal fit with the job [66,67]. When community social workers have a high level of cognitive crafting, they will love the job intrinsically. According to the JD-R theory, the inner passion for work is a kind of individual's resources that plays a buffering role against job demands and behavior. Therefore, community social workers, who can craft the job cognitively, have higher capacity to cope with the pressure caused by job demand. Hence, the effect of role overload on the job burnout will be lower. When community social workers, who can't craft the job cognitively, have lower capacity to cope with the pressure caused by job demand, the effect of role overload on the job burnout will be stronger. Therefore, we propose the following hypotheses:

H3a: Cognitive crafting moderates the relationship between role overload and job burnout, such that the relationship is lower with increasing levels of cognitive crafting.

*H3a: Cognitive crafting moderates the relationship between role overload and deindividuation., such that the relationship is lower with increasing levels of cognitive crafting.*

*H3b: Cognitive crafting moderates the relationship between role overload and emotional exhaustion, such that the relationship is lower with increasing levels of cognitive crafting.*

*H3c: Cognitive crafting moderates the relationship between role overload and personal achievement, such that the relationship is lower with increasing levels of cognitive crafting.*

## 4. Methodology

### 4.1. Sample and data collection

The respondents were mainly community social workers in Suzhou, Nantong, Nanjing and Hangzhou City. Due to the constraints of time, funds, and epidemic prevention and control, the survey was conducted by an online questionnaire. 500 questionnaires were distributed, including the respondents' informed consent, which was approved by UCHRP. 474 questionnaires were collected from September 15 to December 15 in 2021. 462 valid questionnaires were collected after excluding questionnaires with too short response time, vacant questions, and questionnaires with the same options for all items. The study protocol was approved by the ethics review board of Anhui University of Finance and Economics. We have obtained written informed consent from all study participants. All of the procedures were performed in accordance with the Declaration of Helsinki and relevant laws and policies in China.

As can be seen from Table 1, most of the respondents have college and less than a college bachelor's degree. The educational characteristic of the sample is consistent with the proportion of bachelor's degrees in the social worker workforce which is about 65%. The surveyors working between 5 and 10 years occupy the proportion of 40% which almost equals the statistical data of 2021. The proportion of women is much higher than that of men, with the ratio of the two being about 1:3, which is consistent with the ratio of occupational characteristics in national statistics, and thus the sample is representative. The sample profile is shown in Table 1.

**Table 1. Characteristics of the sample.**

| Indicators | Index subdivision dimension | Number of sample | proportion/% |
|---|---|---|---|
| Gender | Male | 113 | 24.5% |
| | Female | 349 | 75.5% |
| Age | Age 30 and under | 91 | 19.7% |
| | 31-40 years old | 224 | 48.5% |
| | 41-50 years old | 121 | 26.2% |
| | Age 50 and above | 26 | 5.6% |
| Education Degree | Junior college and below | 162 | 35.1% |
| | Undergraduate course | 292 | 63.2% |
| | Master's degree or above | 8 | 1.7% |
| Working years | The following 5 years | 127 | 27.5% |
| | 5-10 years | 180 | 39.0% |
| | 11-20years | 131 | 28.4% |
| | More than 20 years | 24 | 5.1% |
| Social Education Major | True | 34 | 7.4% |
| | False | 428 | 92.6% |
| Compensation | 3,000 yuan of the following | 25 | 5.4% |
| | 3,001-4,000 Yuan | 329 | 71.2% |
| | 4,001-5,000 Yuan | 95 | 20.6% |
| | More than 5,000 Yuan | 13 | 2.8% |

## 4.2. Measures

The measures were adapted from well-established items. All measures were assessed with a 5-point Likert scale; one (1) represents strongly disagree, and five (5) represents strongly agree. The items of work-family conflict, which were derived [68], consists of 2 sub-scales with 11 items. The scales of role overload were derived from [69] with 5 items. Cognitive crafting was measured using 6 items from [70]. The scale of job burnout was adopted from [71], which consists of 16 items.

## 4.3. Statistical methods

This data analysis used R-studio software. Firstly, descriptive statistical analysis was conducted to calculate the mean and standard deviation of each variable. Secondly, confirmatory factor analysis was conducted to validate the reliability and validity of the scales. Thirdly, correlation analysis and hierarchical regression were conducted to test the hypothesis.

# 5. Analyses and results

## 5.1. Validity and reliability

Confirmatory factor analysis was conducted to assess the measurement. As shown in Table 2, the CR values are greater than 0.8, and Cronbach α values are all greater than 0.7, indicating acceptable reliability. The AVE values are greater than 0.7, suggesting acceptable convergent validity.

## 5.2. Common method bias test

Harman single factor is used to test the common method bias. All items were loaded into a latent variable, However, the fitting degree of the single factor model is poor: $X^2/DF = 31.67 > 3$, $CFI = 0.424 < 0.9$, $TLI = 0.366 < 0.9$, $RMSEA = 0.258 > 0.08$, $SRMR = 0.19 > 0.06$. The six-factor model has a better fitting degree: $X^2/df = 2.52 < 3$, $CFI = 0.974 > 0.9$,

**Table 2. Measurement model.**

| Variable | Item | Loading | AVE | C.R. | Cronbach's alpha |
|---|---|---|---|---|---|
| Work interferes with family (WIF) | Work interferes with family1 | 0.927 | 0.836 | 0.953 | 0.953 |
| | Work interferes with family2 | 0.917 | | | |
| | Work interferes with family3 | 0.925 | | | |
| | Work interferes with family6 | 0.892 | | | |
| Family interference at work (FIW) | Family interference at work1 | 0.914 | 0.858 | 0.947 | 0.947 |
| | Family interference at work2 | 0.973 | | | |
| | Family interference at work3 | 0.892 | | | |
| Role overload (RO) | Role overload 1 | 0.889 | 0.773 | 0.931 | 0.929 |
| | Role overload 3 | 0.962 | | | |
| | Role overload 4 | 0.964 | | | |
| | Role overload 5 | 0.726 | | | |
| Cognitive crafting (CR) | Cognitive crafting 2 | 0.978 | 0.938 | 0.978 | 0.978 |
| | Cognitive crafting 3 | 0.981 | | | |
| | Cognitive crafting 4 | 0.946 | | | |
| Deindividuation (DE) | Deindividuation 1 | 0.884 | 0.768 | 0.909 | 0.908 |
| | Deindividuation 3 | 0.936 | | | |
| | Deindividuation 4 | 0.812 | | | |
| Emotional exhaustion (EE) | Emotional exhaustion 3 | 0.952 | 0.877 | 0.955 | 0.954 |
| | Emotional exhaustion 4 | 0.952 | | | |
| | Emotional exhaustion 5 | 0.923 | | | |
| Personal achievement (PA) | Personal achievement 2 | 0.816 | 0.715 | 0.909 | 0.907 |
| | Personal achievement 3 | 0.836 | | | |
| | Personal achievement 4 | 0.859 | | | |

TLI = 0.969 > 0.9, RMSEA = 0.057 < 0.08, SRMR = 0.043 < 0.06. Additionally, it can be seen from Table 3 that there are no correlation coefficients bigger than 0.7 [72], which further indicates that the common method bias problem is not serious.

### 5.3. Correlation analysis

Table 3 shows the mean value and correlation of variables.

It can be seen that work interferes with the family correlates significantly with role overload (r = 0.64, p < 0.001). Work interferes with family positively correlated with deindividuation(r = 0.51, p < 0.001), emotional exhaustion (r = 0.68, p < 0.001). There is a significant negative correlation between work interference with family dimension and personal achievement (r = -0.09, p < 0.05). A significant positive correlation exists between family interference with the work and role overload (r = 0.21, p < 0.001). Family interference at work is positively correlated with deindividuation (r = 0.31, P < 0.001), emotional exhaustion(r = 0.29, P < 0.001), and negatively correlates with personal achievement. Role overload positively correlates with emotional exhaustion(r = 0.55, P < 0.001), and deindividuation (r = 0.78, P < 0.001), negatively correlates with personal achievement. This indicates that the significant effect hypothesis of this study is preliminarily supported.

### 5.4. Hypothesis test

Direct effect test. Hierarchical regression is used to test the hypothesis. Table 4 shows that work interference in the family has significant positive effects on deindividuation (M5,

**Table 3. Correlation matrix.**

| | Mean | SD | 1 | 2 | 3 | 4 | 5 | 6 | 7 | 8 | 9 | 10 | 11 |
|---|---|---|---|---|---|---|---|---|---|---|---|---|---|
| Gender | 1.76 | 0.43 | 1.00 | | | | | | | | | | |
| Age | 2.18 | 0.81 | -0.11* | 1.00 | | | | | | | | | |
| Education Level | 1.67 | 0.51 | -0.02 | 0.23*** | 1.00 | | | | | | | | |
| Working years | 2.11 | 0.87 | 0.00 | 0.57*** | -0.07 | 1.00 | | | | | | | |
| Work interferes with family | 4.14 | 0.92 | -0.03 | 0.12** | 0.12** | 0.09* | 1.00 | | | | | | |
| Family interference at work | 2.56 | 1.32 | -0.04 | -0.06 | -0.02 | -0.09 | 0.25*** | 1.00 | | | | | |
| Role overload | 4.12 | 0.86 | -0.09* | 0.19*** | 0.12** | 0.15*** | 0.64*** | 0.21*** | 1.00 | | | | |
| Cognitive crafting | 3.61 | 0.86 | 0.06 | 0.13*** | -0.02 | -0.03 | -0.06 | 0.04 | -0.02 | 1.00 | | | |
| De-individuation | 3.30 | 1.08 | 0.05 | 0.00 | 0.11* | 0.00 | 0.51*** | 0.31*** | 0.55*** | 0.18*** | 1.00 | | |
| Emotional exhaustion | 3.90 | 1.02 | -0.09* | 0.14*** | 0.13** | 0.10* | 0.68*** | 0.29*** | 0.78*** | -0.12** | 0.71*** | 1.00 | |
| Personal achievement | 3.78 | 0.78 | 0.01 | 0.04 | -0.08 | 0.06 | -0.09* | -0.05 | -0.06 | 0.59*** | 0.30*** | 0.16*** | 1.00 |

***P < 0.001, **P < 0.01, *P < 0.05.

β=0.605, P < 0.001), emotional depletion (M13, β=0.734, P < 0.001), and negatively affected personal achievement (M21, β=-0.075). Therefore, H1a was supported. At the same time, family interference in work significantly positively affects deindividuation (M6, β=0.263, P < 0.001), emotional depletion (M14, β=0.233, P < 0.001), and negatively affects personal achievement (M22, β=-0.026). H1b was supported.

Mediation Analysis. Furthermore, we examined the mediating role of role overload between work-family conflict and job burnout. In work-family conflict, the work-family interference (M2, β=0.569, P < 0.001) and the family-interference work (M3, β=0.148, P < 0.001) have significant positive effects on role overload. Role overload positively affects deindividuation (M7, β=0.727, P < 0.01), emotional exhaustion (M15, β=0.925, P < 0.01), and negatively affects personal achievement (M23, β=-0.057). After adding the mediator variable (role overload), role overload had a significant positive effect on deindividuation (M8, β=0.518, P < 0.01). In contrast, the positive effect of the work-interference family on deindividuation (M8, β=0.310, P < 0.01) decreased. Role overload has a significant positive effect on deindividuation (M9, β=0.666, P < 0.01), while family intervention restricts deindividuation (M9, β=0.165, P < 0.01). Role overload has a significant positive effect on emotional failure (M16, β=0.698, P < 0.01), while role overload has a significant positive effect on emotional failure (M16, β=0.337, P < 0.01). Role overload has a significant positive effect on emotional failure (M17, β=0.887, P < 0.01), while the family interference dimension harmed emotional failure (M17, β=0.102, P < 0.01). Role overload restricts personal achievement (M24, β=-0.010), and the negative impact of work interference and family dimension on personal achievement (M24, β=-0.069) increases. Role overload restricts personal achievement (M25, β=-0.050). The negative impact of family interference on personal achievement (M25, β=-0.019) decreased. Therefore, role overload significantly mediates between the work-family conflict and job burnout. Hypothesis H2 is verified.

Moderation Analysis. We first implemented a standardized treatment for role overload and cognitive crafting to examine the moderating effect of role overload and job burnout, to construct interaction terms to exclude the adverse effects of multicollinearity. As shown in Table 4, the moderating effect of cognitive crafting on role overload, depersonalization, and emotional exhaustion is insignificant, while the moderating effect of role overload and reduced personal accomplishment is significant. To further analyze the moderating effect, we plot the moderation figure of role overload versus depersonalization (Fig 2), emotional exhaustion

**Table 4. Hypothesis test.**

| Variables and Models | Role overload | | | | | Deindividuation | | | | | |
|---|---|---|---|---|---|---|---|---|---|---|---|
| | M1 | M2 | M3 | M4 | M5 | M6 | M7 | M8 | M9 | M10 | M11 |
| **trol variables** | | | | | | | | | | | |
| Gender | -0.137 | -0.125 | -0.118 | 0.146 | 0.160 | 0.181 | 0.247* | 0.225* | 0.260** | 0.266** | 0.267** |
| Age | 0.201*** | 0.104* | 0.209*** | 0.053 | -0.049 | 0.067 | -0.093 | -0.104 | -0.072 | -0.142* | -0.142* |
| The degree of education | 0.277*** | 0.114 | 0.290*** | 0.246* | 0.072 | 0.269* | 0.044 | 0.013 | 0.076 | 0.020 | 0.019 |
| Working Years | 0.054 | 0.042 | 0.071 | -0.022 | -0.035 | 0.007 | -0.061 | -0.057 | -0.040 | -0.044 | -0.044 |
| **Independent variables** | | | | | | | | | | | |
| Work interferes with family | | 0.569*** | | | 0.605*** | | | 0.310*** | | | |
| Family interference at work | | | 0.148*** | | 0.263*** | | | | 0.165*** | | |
| **Mediator** | | | | | | | | | | | |
| Role overload | | | | | | | 0.727*** | 0.518*** | 0.666*** | 0.732*** | 0.717*** |
| **Moderator** | | | | | | | | | | | |
| Cognitive crafting | | | | | | | | | | -0.244*** | -0.262 |
| **Interaction** | | | | | | | | | | | |
| Role overload * Cognitive crafting | | | | | | | | | | | 0.004 |
| **R²** | 0.070 | 0.427 | 0.122 | 0.015 | 0.272 | 0.119 | 0.328 | 0.369 | 0.366 | 0.364 | 0.364 |
| **F** | 8.628 | 67.94 | 12.61 | 1.778 | 34.02 | 12.27 | 44.46 | 44.41 | 43.78 | 43.48 | 37.19 |
| **△R²** | | 0.357 | 0.052 | | 0.257 | 0.056 | | 0.041 | 0.038 | | 0.000 |

| Variables and Models | Emotional exhaustion | | | | | | | | |
|---|---|---|---|---|---|---|---|---|---|
| | M12 | M13 | M14 | M15 | M16 | M17 | M18 | M19 | |
| **Control variables** | | | | | | | | | |
| Gender | -0.162 | -0.146 | -0.131 | -0.035 | -0.058 | -0.026 | -0.024 | -0.026 | |
| Age | 0.203** | 0.079 | 0.215** | 0.017 | 0.006 | 0.030 | -0.009 | -0.008 | |
| The degree of education | 0.336*** | 0.125 | 0.356*** | 0.079 | 0.046 | 0.099 | 0.067 | 0.069 | |
| Working years | 0.025 | 0.009 | 0.051 | -0.026 | -0.020 | -0.012 | -0.016 | -0.014 | |
| **Independent variables** | | | | | | | | | |
| Work interferes with family | | 0.734*** | | | 0.337*** | | | | |
| Family interference at work | | | 0.233*** | | | 0.102*** | | | |
| **Mediator** | | | | | | | | | |
| Role overload | | | | 0.925*** | 0.698*** | 0.887*** | 0.928*** | 0.989*** | |
| **Moderator** | | | | | | | | | |
| Cognitive crafting | | | | | | | -0.129*** | -0.057 | |
| **Interaction** | | | | | | | | | |
| Role overload * Cognitive crafting | | | | | | | | -0.018 | |
| **R²** | 0.053 | 0.475 | 0.143 | 0.618 | 0.673 | 0.634 | 0.629 | 0.629 | |
| **F** | 6.382 | 82.37 | 15.24 | 147.3 | 155.7 | 131.3 | 128.6 | 110.1 | |
| **△R²** | | 0.422 | 0.09 | | 0.055 | 0.016 | | 0.000 | |

| | Personal achievement | | | | | | | | |
|---|---|---|---|---|---|---|---|---|---|
| | M20 | M21 | M22 | M23 | M24 | M25 | M26 | M27 | |
| **Control variables** | | | | | | | | | |
| Gender | 0.008 | 0.006 | 0.005 | 0.001 | 0.005 | -0.001 | -0.044 | -0.053 | |
| Age | -0.009 | 0.004 | -0.010 | 0.003 | 0.005 | 0.001 | 0.114 | 0.116 | |
| The degree of education | -0.117 | -0.095 | -0.119 | -0.101 | -0.094 | -0.105 | -0.047 | -0.035 | |
| Working years | 0.053 | 0.054 | 0.050 | 0.056 | 0.055 | 0.053 | 0.016 | 0.245 | |
| **Independent variables** | | | | | | | | | |
| Work interferes with family | | -0.075 | | | -0.069 | | | | |

*(Continued)*

**Table 4.** (Continued)

| Variables and Models | Role overload | | | | | Deindividuation | | | | | |
| --- | --- | --- | --- | --- | --- | --- | --- | --- | --- | --- | --- |
| | M1 | M2 | M3 | M4 | M5 | M6 | M7 | M8 | M9 | M10 | M11 |
| Family interference at work | | | -0.026 | | | -0.019 | | | | | |
| **Mediator** | | | | | | | | | | | |
| Role overload | | | | -0.057 | -0.010 | -0.050 | -0.068 | 0.214* | | | |
| **Moderator** | | | | | | | | | | | |
| Cognitive crafting | | | | | | | 0.554*** | 0.884** | | | |
| **Interaction** | | | | | | | | | | | |
| Role overload * Cognitive crafting | | | | | | | | -0.082*** | | | |
| **R²** | 0.009 | 0.016 | 0.011 | 0.013 | 0.016 | 0.014 | 0.370 | 0.382 | | | |
| **F** | 1.03 | 1.518 | 1.006 | 1.157 | 1.268 | 1.038 | 44.46 | 40.01 | | | |
| △**R²** | | 0.007 | 0.002 | | 0.003 | 0.001 | | 0.012 | | | |

***P < 0.001, **P < 0.01, *P < 0.05.

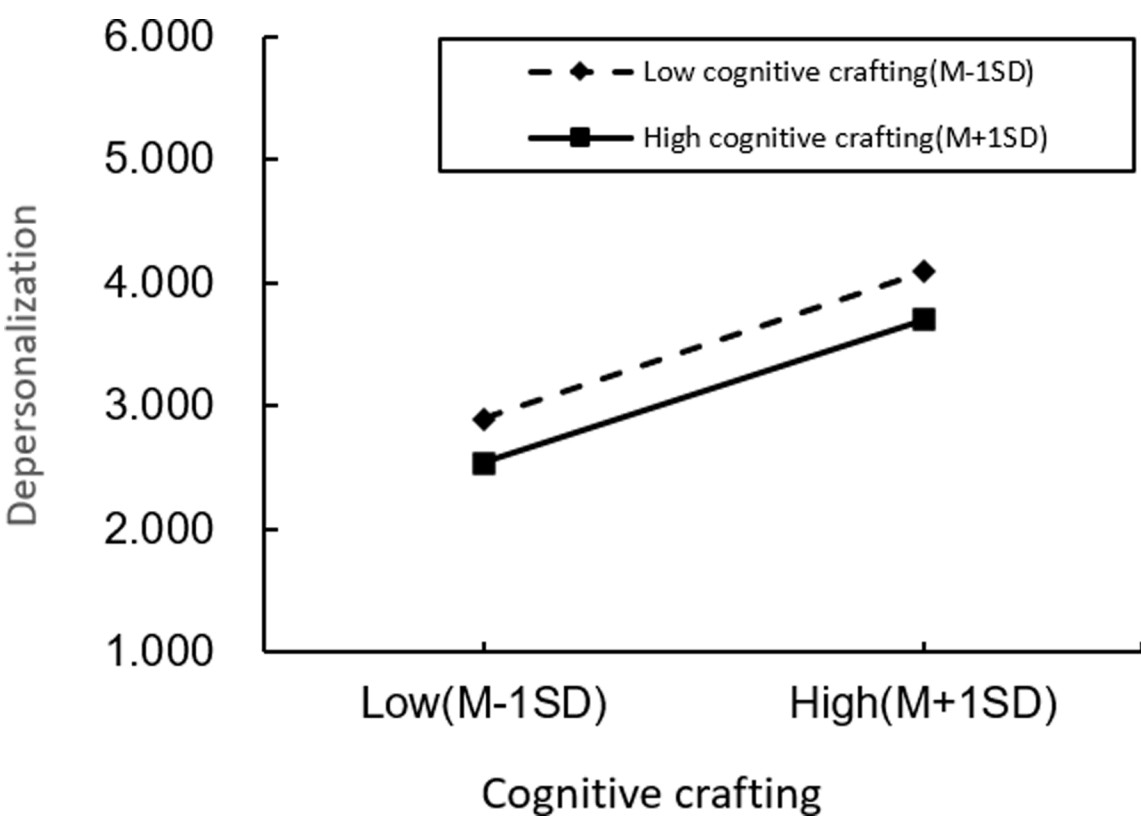

**Fig 2. Cognitive crafting moderation.**

(Fig 3), and the individual are made at high levels of cognitive crafting (a standard deviation above the mean) and low levels of cognitive crafting (a standard deviation below the mean), respectively (Fig 4).The interaction term of role overload and cognitive crafting has a non-significant effect on depersonalization and emotional exhaustion. The relationship between the interaction term of role overload and cognitive crafting and the personal fulfillment of job burnout is significant (M27, β=-0.082, p < 0.001). This suggests that the effects of cognitive

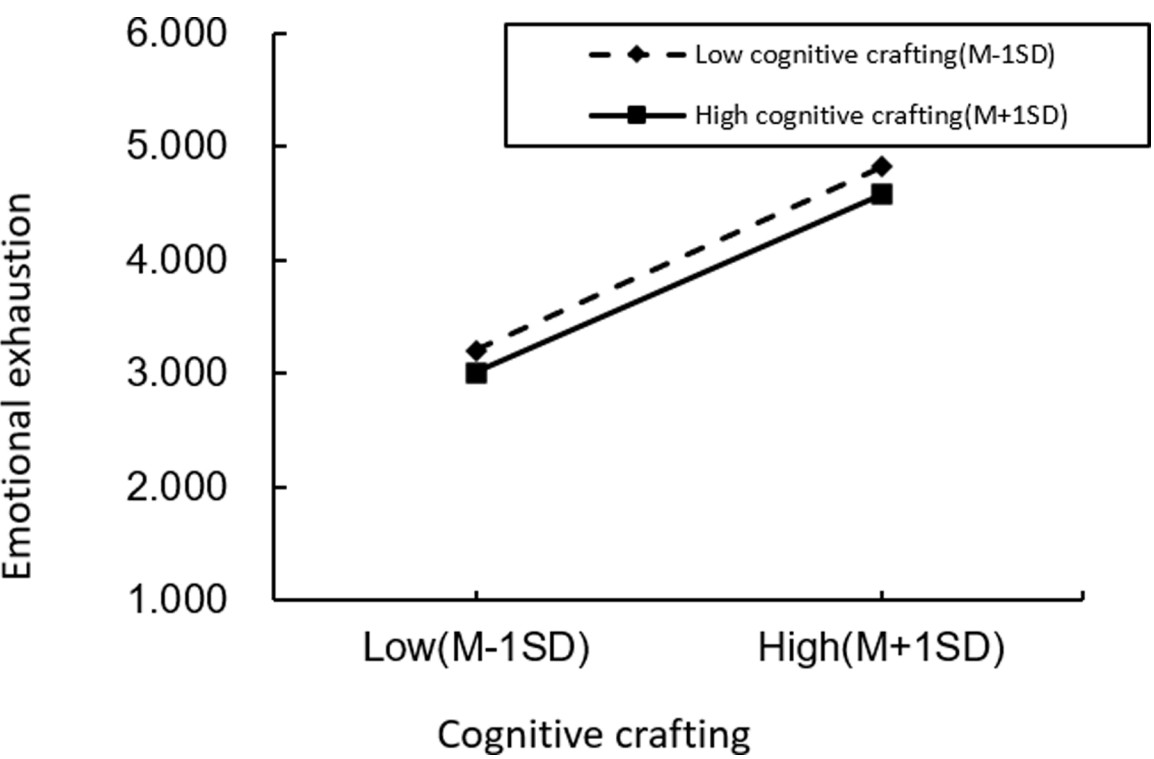

**Fig 3. Cognitive crafting moderation.**

crafting and role overload on depersonalization and emotional exhaustion are independent of each other, with role overload positively predicting depersonalization and emotional exhaustion and cognitive crafting negatively predicting depersonalization and emotional exhaustion and that the failure of cognitive crafting to show significant moderating effects in the effects of role overload and the depersonalization and emotional exhaustion dimensions of job burnout may be restricted by its main effects and may also be influenced by the community workers' personality traits and personality characteristics.

## 6. Findings and implications

### 6.1. Major findings

The main findings are as follows.

First. Work-family conflict affects emotional exhaustion, depersonalization, and personal fulfillment, respectively. This result is consistent with the idea of (Allen, French, Dumani, & Shockley)'. They argued that when work-family conflict is high, individuals can't shoulder the job and family responsibilities simultaneously, they will display negative emotions such as anxiety [58,73]. The findings again confirms that family-work conflict as a destructive work demand negatively affects the health of community workers.

Second. Role overload mediates the relationship between work-family conflict and burnout, which supports our initial hypothesis. The conclusion is identical to prior researchers' arguments [41,74]. They pointed out that role overload leads to an enhanced experience of emotional exhaustion. Therefore, when community social workers are overwhelmed by family and work, role overload can lead to burnout as they do not have the time to engage in additional community work, or even social work, which can lead to burnout.

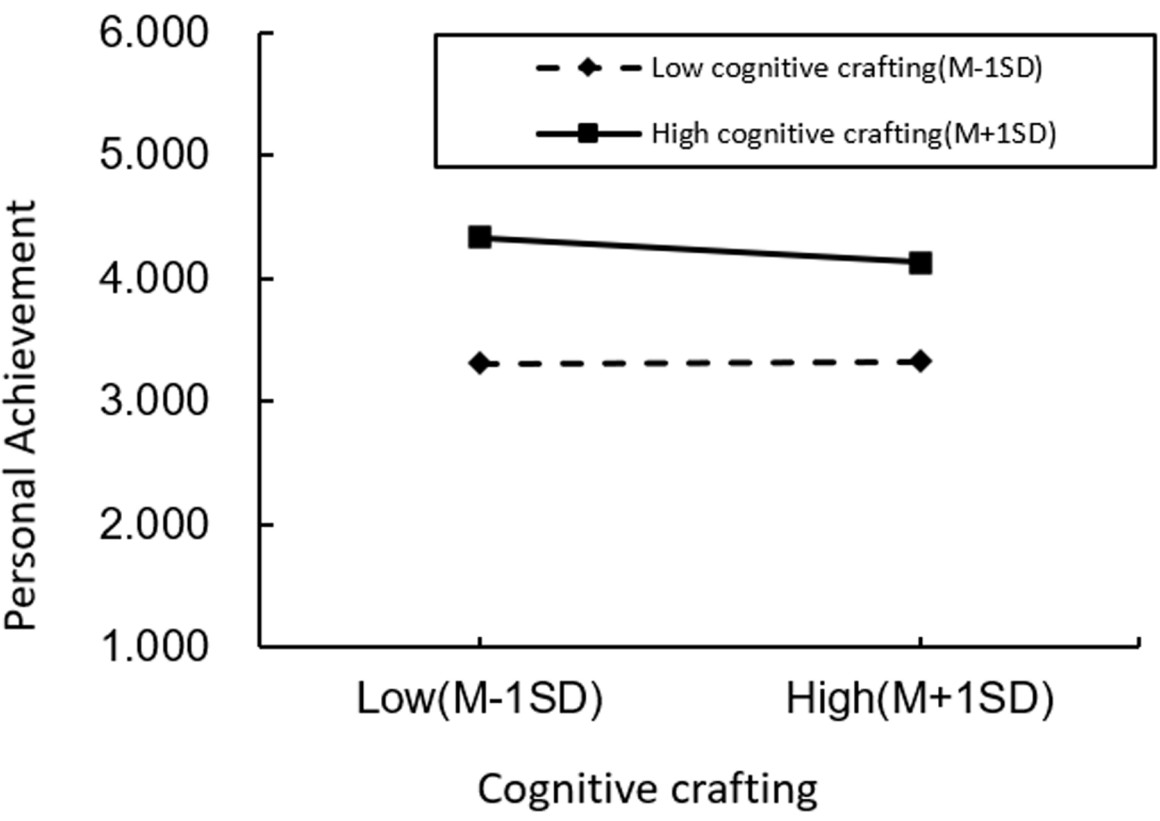

**Fig 4. Cognitive crafting moderation.**

Third. Research has found that cognitive processing moderates role overload and burnout. However, we found job crafting does not moderate the relationship between work-family conflict and job burnout, only partially moderated the relationship between role overload and job burnout. The reason for the inconsistent findings may be that previous studies have not subdivided the dimensions of job crafting and have mainly focused on relational and task crafting [75].

## 6.2. Theoretical implications

This study contributes to the theoretical literature in three ways.

First, the study shows that role overload is an intermediate mechanism of the influence of work-family conflict on job burnout. Cognitive crafting plays an essential role in alleviating job burnout, which can theoretically reveal the internal mechanism of the effect of work-family conflict on job burnout and provide a theoretical basis for effectively preventing or alleviating the management practice of community workers' job burnout. The research introduces cognitive crafting under work challenges as a moderating variable in the JD-R model, which optimizes the theoretical two-path study and provides theoretical evidence on the impact of the interaction between work demands and work resources on job burnout.

Second, most of the existing literature about the effects of job burnout is built from a psychological perspective in Western countries and mainly focuses on healthy social workers [36]. Few research concerns the community social worker in the Chinese context. Our study constructs a model with Chinese characteristics and enriches the localized research on job burnout.

Third, the study responds to the voices in the field of public management, calling for more attention to job burnout and further enriching the research in public behavioral management. The current study of job burnout mainly targets special groups such as healthcare workers, teachers, and police officers [34]. Community social workers are involved in extensive and persistent job burnout behaviors, but their job burnout has been ignored to some extent in the research. This study demonstrates the job burnout states of community social workers, especially during the epidemic period, which may advance the interpretation of employee behavior in public organizations, and remind academics not to ignore emotional labor as an essential phenomenon of the public service field in China.

## 6.3. Managerial implications

This research offers several practical implications.

First, community managers should seek to further clarify roles and responsibilities at work and consider the family situation of community workers to avoid putting them in a state of role ambiguity and overload. IT can be done by scientifically planning, designing job descriptions, setting clear goals for all community workers, reasonably positioning roles, arranging tasks and rewards, valuing support for the family aspects of community workers, allowing employees to use flexible work schedules when necessary, and showing appropriate understanding and concern when taking time off to care for children, the elderly and other family members.

Second, managers should pay attention to the physical and mental health of community workers. Studies have found that employees' negative emotions may trigger behaviors such as job withdrawal, deviation and risk-taking, and have subsequent negative effects on their physical and mental health. There are many ways for organizations to monitor employees' mental health. The organization can increase community workers' knowledge and self-efficacy through training to offset the harm brought by bad emotions, and can also organize psychologists to develop programs to form a three-tier care system for communities, organizations, and families, to promptly assist community workers to channel their psychological problems, reduce their negative feelings and avoid burnout at work in the community.

Third, for the community workers themselves, their own strength and job competitiveness are the first factor to reduce job burnout. Less family burden can make them better face the problems at work, and good working conditions will also contribute to the harmony of the family. Problems shall be adjusted and adapted promptly.

## 6.4. Limitations and future research direction

This study has several limitations too, which could be the direction for future studies. First, the sample size was relatively small in serval cities, and future studies may expand to other cities. Second, although job burnout was examined in this study, given the higher rate of turnover of community social workers, future studies can explore the effect of work-family conflict and job burnout on turnover. Third, a multi-level study could examine more sophisticated interplays of work-family management practices on job burnout.

## Author contributions

**Conceptualization:** Dian Song, Hainan Wu.

**Formal analysis:** Jie Zhao.

**Funding acquisition:** Hainan Wu.

**Investigation:** Jie Zhao.

**Methodology:** Jie Zhao, Hainan Wu.

**Project administration:** Hainan Wu.

**Resources:** Xueyi Ji.

**Software:** Xueyi Ji.

**Supervision:** Hainan Wu.

**Writing – original draft:** Dian Song, Jie Zhao.

**Writing – review & editing:** Hainan Wu, Xueyi Ji.

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
