## [Decision Letter · Decision Letter 0]

9 Jan 2024

PONE-D-23-36476The Impact of Work-family Conflict on job Burnout among Community Workers in ChinaPLOS ONE

Dear Dr. Wu,

Thank you for submitting your manuscript to PLOS ONE. After careful consideration, we feel that it has merit but does not fully meet PLOS ONE’s publication criteria as it currently stands. Therefore, we invite you to submit a revised version of the manuscript that addresses the points raised during the review process.

We look forward to receiving your revised manuscript.

Kind regards,

Zahra Masood Bhutta

Academic Editor

PLOS ONE

Journal Requirements:

"This study is a phased achievement of the  Humanities and Social Sciences Key Project in Anhui Province (2022AH050570), funded by the Anhui Provincial Department of Education."

"The authors acknowledge financial support from Humanities and Social Sciences Project in Anhui Universities (2022AH050570). This paper has been continuously revised in the light of comments of the anonymous reviewers and referee, to whom the authors are thankful. The usual disclaimer, however, applies."

"This study is a phased achievement of the  Humanities and Social Sciences Key Project in Anhui Province (2022AH050570), funded by the Anhui Provincial Department of Education."

Reviewers' comments:

Reviewer's Responses to Questions

**Comments to the Author**

1. Is the manuscript technically sound, and do the data support the conclusions?

Reviewer #1: Yes

Reviewer #2: Partly

2. Has the statistical analysis been performed appropriately and rigorously? 

Reviewer #1: Yes

Reviewer #2: Yes

3. Have the authors made all data underlying the findings in their manuscript fully available?

Reviewer #1: Yes

Reviewer #2: No

4. Is the manuscript presented in an intelligible fashion and written in standard English?

Reviewer #1: Yes

Reviewer #2: Yes

5. Review Comments to the Author

Reviewer #1: The paper is well structured and well written besides focusing on a relevant and timely topic and merits publication subject to incorporating changes/suggestions on a few areas that need further improvement. These are as under;

1- The Introduction section needs to develop proper rational and motivation for the study. For example why not building the rational of the study in context of COVID-19.

2- Through out the document and specially in the Introduction Section, the author has made claims that need to be properly backed by citations. For example we can see phrases in the intro like " Many researchers have explored...." , "Much has been explored by researchers" , and " Some Scholars have realized that...." are a few to mention here. All these phrases need to be backed by relevant citations.

3-It seems like some of the author guidelines for formatting have been ignored/not followed in some places. For example we can see spaces before headings at some places and even the way Citations are done in text in the literature section are not in the standard format.

4-Most of the literature cited in the text of the document is outdated. More relevant and fresh studies need to be cited. Specially in the Literature Section. Moreover, reference to studies conducted in the Chinese context are highly missing and must be included.

5-In Section 2.5 i.e Job Demand-Resource Model, the phrase "...formed by two kind of factors and the health....." is too long and also seems incomplete and hence conveys no meaning.

6-The Literature Review section needs to be split into two distinct parts i.e., i) Empirical Literature and ii) Theoretical Literature/Underpinnings. Currently the theoretical literature citations and theoretical underpinnings are merged which creates confusion.

7-The diagram "Theoretical Framework" seems miss-labelled. It should be re-labelled as "Conceptual Model".

8-All the Hypotheses need to be backed/supported with the help of relevant literature. Currently in the document this section does not have any citations which is strange to note.

9-The author needs to stick to one type of phrasing the hypotheses. The use of words "in other words" is not appropriate. Moreover, the versions of hypotheses mentioned after "in other words" seem more appropriate and therefore only those should be mentioned.

10-The word " Samples" in the heading of Section 4.1 and Table 1 needs to be corrected. It should be "Sample".

11-While the Author has mentioned details of the number of respondents and questionnaires returned etc. We still fail to see details of the figures about i) the population ii) No of questionnaires that were actually distributed/sent/shared iii) The sample selection criteria (eg any formula or procedure followed to arrive at the current number of respondents)

12-A very striking remark is that the Table 1 on Characteristics of the Sample shows that about 75.5% of the sample consists of Female respondents. The author has even also commented on it. However, it seems that this thing is making the sample highly biased towards females only. The Author needs to build justification of this bias or present a proper rationale behind focusing more on females.

13-Finally, while the author has followed a very rigorous statistical procedure and has derived a number of findings, a comparison with similar/dissimilar findings n previous literature with citations needs to be incorporated to improve the quality of the discussion on findings.

Reviewer #2: Overall, It was a good effort. I have few suggestions which may include: Rewrite the abstract as it was not depicting the whole article, add latest references in introduction section, add few lines to describe the importance of the research, research problem should be backed with contemporary references, the researchers may add a paragraph in methodology section to describe the data collection methods, sampling technique and data analysis method, tables should be presented in an appropriate way, researcher should add last five years references in all chapters.

6. PLOS authors have the option to publish the peer review history of their article (what does this mean? ). If published, this will include your full peer review and any attached files.

**Do you want your identity to be public for this peer review?** For information about this choice, including consent withdrawal, please see our Privacy Policy .

Reviewer #1: **Yes: ** Dr. Majid Jamal Khan

Reviewer #2: **Yes: ** Muhammad Faizan Khan

---

## [Author Response · Author response to Decision Letter 1]

1 Mar 2024

Dear Editors,

We greatly appreciate the opportunity to revise our manuscript entitled " The Impact of Work-Family Conflict on Job Burnout among Community Social Workers in China". The thoughtful guidance and insightful advice provided by you and the reviewers have helped us improve the quality of the manuscript. We have made efforts to take action to address all issues by reviewers in the revision. The modifications made to every comment are presented in detail below.

Thank you again for all the insightful guidance and encouraging comments. We hope you and the reviewers will find our revision satisfactory for further consideration.

Best regards,

Authors.

---

## [Decision Letter · Decision Letter 1]

20 Mar 2024

The Impact of Work-family Conflict on job Burnout among Community Workers in China

PONE-D-23-36476R1

Dear Dr. Wu,

We’re pleased to inform you that your manuscript has been judged scientifically suitable for publication and will be formally accepted for publication once it meets all outstanding technical requirements.

An invoice for payment will follow shortly after the formal acceptance. To ensure an efficient process, please log into Editorial Manager at Editorial Manager® , click the 'Update My Information' link at the top of the page, and double check that your user information is up-to-date. If you have any billing related questions, please contact our Author Billing department directly at authorbilling@plos.org.

Kind regards,

Zahra Masood Bhutta

Academic Editor

PLOS ONE

---

## [Editor Report · Acceptance letter]

PONE-D-23-36476R1

PLOS ONE

Dear Dr. Wu,

I'm pleased to inform you that your manuscript has been deemed suitable for publication in PLOS ONE. Congratulations! Your manuscript is now being handed over to our production team.

Kind regards,

on behalf of

Dr. Zahra Masood Bhutta

Academic Editor

PLOS ONE